# From Code Review to Spec-Driven Contracts:
# A Vision for Auditable AIWare Systems

## Abstract

AI-driven software systems are increasingly developed through rapid, iterative practices that combine large language models, prompt engineering, and ad-hoc integration of external tools and services; a style often described as *vibe coding*. While these practices enable fast experimentation and deployment, they challenge the basic principles of software engineering. Documentation is informal and quickly outdated, requirements are often implicit, and code review and testing are applied to artifacts that only partially determine system behavior. As a result, critical questions about whether a system behaved as intended, permitted, or prohibited cannot be reliably answered after deployment. This paper presents a vision for *spec-driven, contract-based AIWare systems* in which specifications function as explicit communicative commitments defining *required*, *permitted*, and *forbidden* behavior. We argue that auditability cannot be achieved through code review alone, and instead requires specifications that are enforceable across continuous integration and deployment (CI/CD) pipelines, runtime execution, and post-hoc audit. We introduce a contract-driven framework structured around specification, execution, and audit planes, extend it with spec-driven CI/CD integration, and illustrate the approach through walkthrough examples. Our vision reframes auditability as a first-class system property and specifications as the authoritative source of correctness in AIWare systems.

## ACM Reference Format:
Anonymous Author(s). 2026. From Code Review to Spec-Driven Contracts: A Vision for Auditable AIWare Systems. In *Proceedings of The 3rd ACM International Conference on AI-powered Software (AIware 2026)*. ACM, New York, NY, USA, 5 pages. https://doi.org/10.1145/nnnnnnn.nnnnnnn

## 1 Introduction

An AIware system is software whose behavior is built by orchestrating AI components, often large language models, together with code, external tools, services, and runtime context, and engineered using AI-native software engineering practices [1]. In practice, AIWare development is fast, iterative, and highly dynamic: prompts evolve, models are updated, tools are added or removed, and system behavior shifts without corresponding changes to application code [2].

These practices enable rapid experimentation and deployment, but they strain foundational assumptions of software engineering.

Traditional workflows assume that system behavior can be understood, validated, and trusted through static artifacts such as source code, tests, and documentation [3]. Code review plays a central role in this model as the primary mechanism for assessing correctness prior to deployment [3].

AIWare systems challenge this premise, as their behavior is not fully determined by code alone but emerges from interactions among prompts, probabilistic model inference, orchestration logic, runtime context, and external services [4, 5]. Empirical evidence shows that small changes, such as prompt edits or configuration adjustments, can lead to qualitatively different behavior that is not always detected by existing testing practices [4]. Consequently, failures often stem not from faulty implementations, but from mismatches between intended behavior and observed execution at runtime [6].

This shift exposes a deeper problem: existing practices are poorly equipped to answer a critical post-deployment question; *did the system behave as it was allowed to behave?* When unexpected behavior occurs, developers and reviewers must reconstruct intent retrospectively by interpreting documentation, code diffs, test coverage, or runtime logs [3, 7]. While such practices may help explain how behavior arose, they cannot reliably determine whether that behavior was required, permitted, or forbidden at the time of execution [6].

We argue that *auditability* is the missing system property in AIWare engineering. We define auditability as the ability to verify, after execution and without reliance on subjective interpretation, whether a system's behavior conformed to explicit commitments. Auditability is not synonymous with testing, monitoring, or observability: tests indicate what was exercised, logs record what happened, and reviews assess plausibility, but none establish normative compliance. Achieving auditability requires a shift in how correctness is defined and enforced. Rather than treating specifications as descriptive documentation or design-time guidance, AIWare systems must treat specifications as explicit, enforceable commitments that define acceptable behavior. These commitments must persist across delivery pipelines, runtime execution, and post-hoc analysis; without them, correctness remains a matter of interpretation rather than verification [8, 9].

This paper presents a vision for *spec-driven, contract-based AIWare systems*. In this vision, specifications function as authoritative communicative commitments that declare required, permitted, and forbidden behavior [10]. These commitments are enforced across CI/CD pipelines and runtime execution and ground the production of audit evidence [6, 9]. We introduce a contract-driven framework and show how it prevents failures that pass traditional review and testing, shifting AIWare engineering from inferring intent to demonstrating compliance.

## 2 Limits of Existing Practices

Software engineering relies on documentation, requirements specification, and code review to establish confidence in system behavior [3]. These practices remain essential for coordination, design reasoning, and quality control. However, even when applied rigorously, they are structurally ill-suited to support *auditability* in AIWare systems [6]. The core limitation is not poor execution of these practices, but a mismatch between their fundamentally interpretive nature and the need for post-hoc, non-interpretive verification of behavior. Auditability requires the ability to determine whether a system's execution conformed to explicit obligations. Existing practices were not designed to answer that question.

### 2.1 Documentation, Requirements, and Code Review

Documentation is the most common mechanism for communicating system intent. Design documents, READMEs, and comments describe expected behavior and explain how components are intended to interact. However, documentation is fundamentally descriptive rather than normative [11]. It describes typical or intended behavior, but it does not establish binding obligations, nor is it evaluated against runtime execution [11]. In AIWare systems driven by prompt and configuration changes, documentation also diverges quickly from observed behavior.[4]

Requirements engineering introduces explicitly normative language of what a system must or must not do [11], and is therefore closer to the needs of auditability. In practice, however, requirements often lose authority after design time. They are validated during initial development, enforced indirectly through tests or developer discipline, and rarely linked systematically to runtime behavior [11]. Changes to prompts, models, or configuration frequently bypass requirement checks entirely, leaving requirements aspirational rather than binding [4].

Code review is widely treated as the primary quality gate in modern development [3]. Reviewers assess proposed changes for correctness, design quality, and maintainability. However, code review is inherently a pre-execution and interpretive practice [3]. It assumes that relevant behavior is visible in the artifacts under review. In AIWare systems, this assumption often fails: critical behavior may depend on prompt phrasing, probabilistic model inference, tool responses, or runtime context unavailable at review time [5]. As a result, code review can establish whether a change appears reasonable, but not whether executions will conform to behavioral obligations. It evaluates plausibility rather than compliance.

These limitations become clear when contrasted with the requirements of auditability. Code review and auditing answer different questions, at different times, using different forms of evidence [3, 6]. A change may pass review and testing while enabling behavior that violates expectations at runtime, or behavior may change without triggering review at all [12]. Code review therefore cannot serve as a reliable audit mechanism.

Across documentation, requirements, and review, a common assumption persists: that intent can be reconstructed after the fact [3, 13]. When behavior is unexpected, stakeholders attempt to infer what the system was supposed to do by interpreting artifacts and discussions. In AIWare systems, where intent is often indirect and underspecified, this assumption breaks down. Disagreements about correctness cannot be resolved objectively without authoritative commitments [6]. Auditability requires that intent be made explicit *before* execution, not inferred afterward.

### 2.2 From Description to Communicative Commitments

The limitations of existing practices point to a deeper issue: in AIWare, system intent is frequently communicated, but rarely committed. Expectations about acceptable behavior are described in documentation, implied through prompts, or inferred during review [14]; yet they are not expressed in a form that establishes binding obligations over execution.

To support auditability, specifications must be reframed not as descriptive artifacts, but as *communicative commitments*. A commitment is a statement that creates an obligation: it declares what behavior is *required*, *permitted*, or *forbidden*, and establishes accountability if that declaration is violated. This reframing shifts correctness from an interpretive judgment to a verifiable property of system execution.

Documentation answers the question: *"What is this system intended to do?"* Commitments answer a different question: *"What behavior is this system obligated to uphold?"*

This distinction is not rhetorical. Descriptive artifacts tolerate ambiguity and multiple interpretations. Commitments cannot. They must be precise enough to support consistent evaluation against execution. In AIWare systems, where behavior is emergent and context-dependent, interpretive understanding is insufficient. For specifications to function as commitments, they must explicitly answer a set of normative questions:

| Normative Questions for Auditability |
| --- |
| *What behavior is required?*
*What behavior is permitted?*
*What behavior is forbidden?*
*Under what conditions do these norms apply?* |

These questions define the space of acceptable behavior. When answered, they establish a shared reference against which system behavior can be evaluated during delivery, execution, and audit.

Commitments constrain *behavior*, not implementation [10]. They do not prescribe algorithms, model architectures, or prompt structure. This allows specifications to remain stable even as models, tools, and system architecture evolve.

Reframing specifications as commitments also establishes a clear authority hierarchy. Specifications define correctness. Code implements mechanisms. Tests and analyses provide evidence. None of these artifacts supersede the specification. Without authoritative commitments, enforcement logic lacks grounding and audit records lack evaluative meaning.

Together, these observations motivate a contract-driven approach, where specifications serve as the authoritative basis for defining and verifying acceptable behavior. The next section introduces a framework that operationalizes these commitments by separating specification, enforcement, and evidence.

**Table 1: Spec-Driven Contract Framework for AIWare Across the System Lifecycle**

| Plane | Purpose | Defines | Enforces | Does *Not* Determine | Lifecycle Scope |
|---|---|---|---|---|---|
| **Specification Plane** | Declare authoritative commitments | Obligations (REQUIRED), permissions (PERMITTED), prohibitions (FORBIDDEN), and conditions over behavior | — | Enforcement mechanisms, roles, implementation details, or evidence | Design time; referenced throughout CI/CD and runtime |
| **Execution Plane** | Enforce commitments at points of action | Enforcement responsibilities (e.g., mediating or constraining actions via checks against declared commitments, including role-scoped commitments) and mediation points (e.g., tool or service interactions) | Commitments from the specification plane | New obligations, permissions, prohibitions, or discretionary interpretation of commitments | CI/CD pipelines and runtime execution |
| **Audit Plane** | Enable post-hoc verification and accountability | Evidence schemas and recording obligations linking actions to active commitments and context | — | Normative judgment, inferred intent, or reconstructed obligations | CI/CD artifacts and runtime records |

## 3 Spec-Driven Contracts for AIWare

Reframing specifications as communicative commitments raises a practical question: *how can such commitments be enforced and evaluated in systems whose behavior emerges dynamically at runtime?* Auditability cannot be achieved through specifications alone. It requires mechanisms that carry declared obligations across delivery pipelines, runtime execution, and post-hoc analysis.

Table 1 summarizes our contract-driven approach, in which specifications function as authoritative contracts and are enforced throughout the system lifecycle, including CI/CD pipelines and runtime execution. While prior work has explored contracts, policies, and runtime enforcement, these mechanisms are typically framed as design-time constraints or safety checks [15]. We instead reframe specifications as communicative commitments whose purpose is to enable post-hoc verification that observed behavior was permitted.

### 3.1 A Contract-Driven Framework

Our approach distinguishes three conceptual planes that separate intent, behavior, and evidence: a *specification*, an *execution*, and an *audit plane*. This separation is intentional. Inferring obligations from code or reconstructing intent from logs reintroduces interpretation and undermines auditability.

The *specification plane* captures contractual commitments as explicit obligations, permissions, and prohibitions over system behavior. These commitments define what behavior must, may, or must not occur and serve as the authoritative reference for correctness. Importantly, they are independent of implementation details such as model choice, prompt structure, or orchestration logic, allowing internal mechanisms to evolve without redefining correctness.

The *execution plane* is responsible for enforcing active commitments during runtime. Rather than relying on developer intent or reviewer interpretation, this plane *constrains behavior* at the points *where actions occur*, such as tool invocation or external interaction. Enforcement may include blocking prohibited actions, mediating access to resources, or detecting unmet obligations. Enforcement applies uniformly regardless of whether an action is initiated by an AI component, an automated agent, or a human operator. When contractual commitments are *scoped to roles*, the execution plane enforces them with respect to the active role at the time of action. While the execution plane enforces commitments, it does not define them; authority remains with the specification plane.

The *audit plane* records verifiable evidence of system behavior in relation to active commitments. Audit records capture *what actions occurred*, *under what conditions*, and *which obligations* applied at *the time*. Such records enable post-hoc verification of compliance, accountability, and diagnosis of violations. Without explicit commitments, logs alone cannot establish whether observed behavior was acceptable.

### 3.2 Enforcement Across CI/CD and Runtime

For specifications to function as authoritative contracts, they must be enforced throughout the system lifecycle. CI/CD pipelines provide an early enforcement point, while runtime mechanisms ensure continuity after deployment.

In a spec-driven workflow, CI/CD pipelines act as normative gatekeepers. Rather than asking only whether builds succeed or tests pass, the pipeline evaluates whether proposed changes comply with active contractual commitments. Contract validation ensures that commitments are explicit and internally consistent, while compliance checks detect changes that weaken or bypass existing obligations. Violations are reported as contractual failures, even when all tests pass.

CI/CD execution also produces artifacts that link specifications to enforcement checks and validation outcomes. These artifacts form part of the audit trail and enable downstream verification that deployed behavior corresponds to declared commitments.

CI/CD enforcement is necessary but not sufficient. Some violations can only be observed at runtime due to partial observability, external tool behavior, or context-dependent execution. After deployment, runtime enforcement mechanisms continue to mediate actions with external effects. When enforcement is not possible, attempted or potential violations are recorded together with the active commitments and execution context, preserving auditability across the full lifecycle.

## 4 Walkthrough Example

To illustrate why spec-driven contracts are necessary for auditable AIWare systems, we consider a simplified representative scenario. The purpose is not to present a complete implementation, but to show how violations arise under conventional workflows and how a contract-driven approach changes the outcome.

### 4.1 Scenario

Consider an AI assistant that helps users schedule meetings and send emails. The assistant relies on an LLM to interpret user requests and invokes external tools for calendar access and email

delivery. The system operates in a production environment and performs actions with real-world effects.

## 4.2 Contractual Commitments

The system is governed by the following explicit communicative commitments:

(1) **Confirmation obligation:** The system must not send emails without explicit user confirmation.
(2) **Restricted tool use:** Calendar access is permitted only for scheduling purposes.
(3) **Logging requirement:** All external communications must be logged.

These commitments define acceptable behavior independently of implementation details.

## 4.3 Failure Under a Traditional Workflow

A change is proposed to improve responsiveness by modifying prompt phrasing to reduce explicit confirmation steps in common cases. The code change is minimal and localized.

Under a traditional workflow, code review verifies that confirmation logic still exists and automated tests pass for standard interaction paths. No violations are detected prior to deployment. However, after deployment, the updated prompt causes the LLM to implicitly confirm email-sending actions in certain contexts. In addition, one tool invocation path bypasses the logging mechanism. No application code enforcing these constraints was modified, and no tests failed. As a result, the system sends emails without explicit user confirmation and without producing audit logs. The deviation is silent and is discovered only after external effects occur.

## 4.4 Spec-Driven CI/CD and Runtime Enforcement

Under a spec-driven workflow, the same change enters a CI/CD pipeline where specifications act as normative gatekeepers. Contract validation detects that the change weakens enforcement of the confirmation obligation, and compliance verification identifies a potential execution path that allows email transmission without an explicit confirmation event. The pipeline fails with a contractual violation, blocking deployment despite passing all tests.

If a compliant version is deployed, runtime enforcement mechanisms prevent unauthorized email-sending actions. When enforcement is not possible, attempted violations are recorded along with the active commitments and execution context, producing verifiable audit evidence.

## 4.5 Outcome

This example demonstrates that code review and testing can pass even when contractual obligations are violated at runtime, and that changes outside the application code itself can still introduce unacceptable behavior. It highlights the value of spec-driven CI/CD pipelines in preventing such violations before deployment, and shows how runtime enforcement combined with audit evidence completes the accountability loop.

## 5 Implications and Research Directions

Treating specifications as communicative commitments and enforcing them across CI/CD pipelines and runtime execution has direct implications for both practice and research. Rather than relying on interpretive judgment to establish correctness, spec-driven AIWare systems enable demonstrable compliance with explicit behavioral obligations.

## 5.1 Implications for Practice

In a contract-driven workflow, established software engineering practices are repositioned rather than replaced. Code review remains essential for assessing maintainability, performance, and design quality, but it no longer serves as the primary determinant of correctness. Questions about whether behavior is allowed or forbidden are resolved by reference to explicit commitments, not reviewer inference.

CI/CD pipelines assume a normative role as consistent gatekeepers that prevent non-compliant changes from reaching deployment. Runtime enforcement and audit artifacts provide concrete evidence of compliance or violation, reducing ambiguity during incident response and post-hoc analysis. Together, these shifts replace implicit trust with explicit accountability and better align development workflows with AIWare systems whose behavior cannot be fully anticipated at design time.

## 5.2 Research Directions

Realizing auditable AIWare systems raises several open research challenges:

- Designing specifications that are both human-authorable and machine-enforceable
- Defining governance models for specification ownership and evolution
- Reconciling probabilistic LLM behavior with deterministic contractual obligations
- Detecting semantic drift between specified and observed behavior
- Building transparent and explainable enforcement mechanisms

Addressing these challenges requires advances across requirements engineering, software architecture, development tooling, and socio-technical governance.

## 5.3 Conclusion

AIWare systems expose a growing gap between how software behaves in practice and how correctness and accountability are established. This paper argued that auditability must be treated as a first-class system property, and that achieving it requires treating specifications as enforceable commitments rather than descriptive artifacts.

By introducing a contract-driven approach spanning specification, execution, and audit, and by integrating enforcement across CI/CD pipelines and runtime execution, we showed how AIWare systems can move from inferred intent to verifiable compliance. This shift is essential for building AI-driven systems that are not only capable, but also accountable and worthy of trust.

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
