# OpenReview forum: "From Code Review to Spec-Driven Contracts: A Vision for Auditable AIWare Systems"
_ACM.org/AIWare/2026/Conference — AIware 2026_

### Official Review · Reviewer_7D1k · 2026-03-02

**Rating:** 3
**Confidence:** 4

**Review:**

Strengths
- Clear articulation of the auditability gap in AI-centric systems.
- Strong conceptual distinction between descriptive documentation and binding commitments.
- Clean architectural separation of specification, enforcement, and evidence.
- Timely and relevant to LLM-based system development.

Weaknesses
- No empirical validation or feasibility evaluation.
- Vision-level contribution with no formal model or prototype.
- Enforcement mechanisms remain abstract.
- Limited comparison with design-by-contract and runtime verification literature.

**Summary:**

The paper argues that traditional software engineering practices, particularly code review and testing, are insufficient for AIWare systems whose behavior emerges from prompts, LLM inference, orchestration logic, and runtime context. It introduces auditability as a first-class system property, defined as the ability to verify post-execution whether behavior conformed to explicit commitments.
To address this gap, the authors propose a spec-driven, contract-based framework with three planes:
- Specification plane: Declares REQUIRED, PERMITTED, and FORBIDDEN behavior.
- Execution plane: Enforces commitments in CI/CD and at runtime.
- Audit plane: Records evidence linking behavior to active commitments.
A scheduling assistant example illustrates how prompt-level changes can violate obligations despite passing review and tests, and how spec-driven enforcement prevents such failures

There already has been several related work that are missing from discussion:
[1] Wagner, Stefan, et al. "Towards evaluation guidelines for empirical studies involving llms." 2025 IEEE/ACM International Workshop on Methodological Issues with Empirical Studies in Software Engineering (WSESE). IEEE, 2025.
[2] Baltes, Sebastian, et al. "Guidelines for empirical studies in software engineering involving large language models." arXiv preprint arXiv:2508.15503 (2025).
[3] Ralph, Paul, et al. "Empirical standards for software engineering research." arXiv preprint arXiv:2010.03525 (2020).

---

### Official Review · Reviewer_mXqp · 2026-03-05

**Rating:** 4
**Confidence:** 4

**Review:**

STRENGHTS
+ Very interesting and well-presented idea
+ Very relevant for the AIWare community

WEAKNESSES
- Little discussion of what already exists for ensuring LLMs correctness


DETAILED COMMENTS

I agree with and very much appreciate the intuition behind the paper, namely the idea of enforcing correctness in AIware, which I believe will become a key role of software engineering in the emerging agentic era.

As I understand it, the core idea is to introduce guardrails for agents operating within software ecosystems. I again appreciate the emphasis on auditability. However, since there is already work on guardrails coming from the AI community (e.g., [1]), I think the paper would benefit from discussing such approaches and clarifying why auditability is needed in addition to them, and what additional value it brings.

I particularly like the idea of the three planes, which I believe are very well designed and clearly presented. I also find the discussion on CI/CD integration valuable. One suggestion would be to expand the discussion beyond purely technical settings. In particular, the framework could become even more interesting when considering environments where multiple technicians and stakeholders interact (e.g., DevOps or LLMOps settings). In such contexts, contracts could serve not only as enforcement mechanisms but also as collaborative artifacts: (1) they could help stakeholders understand each other’s intentions and inform developers about expected behaviors, and (2) they could allow specialized professionals to engineer specific requirements—especially non-functional ones such as fairness or security—without necessarily requiring direct interaction with those implementing the system. In this sense, the proposal could also be framed as a socio-technical contribution, rather than only a technical one.

Related to this point, the paper could potentially include a short discussion on qualitative and non-functional aspects. For instance, could the proposed approach support security enhancement? Could it help models produce outputs that are fairer, more secure, more sustainable, or simply more readable?

Another aspect that could be expanded is the technical feasibility of the proposed contract mechanism. In particular, it would be useful to better understand how the authors envision implementing and integrating such contracts into real code repositories, either in industrial or open-source contexts. How easy would it be to adopt in practice? Would this be implemented purely through system prompts, or could it also take the form of explicit pre- and post-condition constraints (for example through a declarative language or DSL)?

Overall, the paper is well written and clearly presented. The topic is also timely and novel within the software engineering community, and I believe it could stimulate valuable discussion and insights if presented and debated within the community.



[1] Traian Rebedea, Razvan Dinu, Makesh Narsimhan Sreedhar, Christopher Parisien, and Jonathan Cohen. 2023. NeMo Guardrails: A Toolkit for Controllable and Safe LLM Applications with Programmable Rails. In Proceedings of the 2023 Conference on Empirical Methods in Natural Language Processing: System Demonstrations, pages 431–445, Singapore. Association for Computational Linguistics

**Summary:**

SUMMARY

The paper proposes a vision for specification-driven AIWare systems aimed at improving the auditability and correctness of AI-driven software developed using practices such as prompt engineering and rapid LLM integration. The framework is organized into three planes (specification, execution, and audit) and is designed to integrate with CI/CD pipelines. The paper presents this approach as a way to make auditability a core property of AI-driven software systems.

---

### Official Review · Reviewer_X71j · 2026-03-06

**Rating:** 3
**Confidence:** 3

**Review:**

The vision seems timely, as verification of system behavior will become more difficult with the rise of AIWare systems. Using specifications that can be checked against seems like a good path forward. While the paper outlines the difference to several existing practices, I'm not clear on why requirements cannot be used: Sec. 2.1 mentions that the problem is that requirements "lose authority" after design time - wouldn't another possibility be to envision a redefined process where requirements are used in later stages as well, instead of introducing additional artifacts to the process? I'm also wondering what the difference to design-by-contract is?

I'm also missing some detail regarding the envisioned implementation of the vision (and, consequently, the feasibility of implementing it): Section 4.4 mentions that the "contract validation detects that the change weakens enforcement [...]" as part of a CI/CD pipeline, but how does it do so if tests do not catch the change? The same section also states that "if a compliant version is deployed, runtime enforcement mechanisms prevent unauthorized email-sending actions" - if a version is "compliant", why does it still require runtime enforcement? What is the difference in system behavior that can be observed/enforced, and how does it translate to the example?

**Summary:**

The paper addresses the problem that it is hard to answer questions regarding the correctness of AIWare systems behavior, as auditability is missing. The vision presented is to use specifications as commitments regarding the behavior, more specifically defining required, permitted, and forbidden behavior. These specifications should then be enforceable across CI/CD and runtime execution, and enable post-hoc auditability. The paper proposes a contract-driven framework for this and gives a walkthrough example.